# Comprehensive mutagenesis maps the effect of all single-codon mutations in the AAV2 *rep* gene on AAV production

Nina K Jain[1,2], Pierce J Ogden[1,2]*[†], George M Church[1,2]*

[1]Wyss Institute for Biologically Inspired Engineering, Boston, United States;
[2]Department of Genetics, Harvard Medical School, Boston, United States

**Abstract** Recombinant adeno-associated viruses (rAAVs) are the predominant gene therapy vector. Several rAAV vectored therapies have achieved regulatory approval, but production of sufficient rAAV quantities remains difficult. The AAV Rep proteins, which are essential for genome replication and packaging, represent a promising engineering target for improvement of rAAV production but remain underexplored. To gain a comprehensive understanding of the Rep proteins and their mutational landscape, we assayed the effects of all 39,297 possible single-codon mutations to the AAV2 *rep* gene on AAV2 production. Most beneficial variants are not observed in nature, indicating that improved production may require synthetic mutations. Additionally, the effects of AAV2 *rep* mutations were largely consistent across capsid serotypes, suggesting that production benefits are capsid independent. Our results provide a detailed sequence-to-function map that enhances our understanding of Rep protein function and lays the groundwork for Rep engineering and enhancement of large-scale gene therapy production.

*For correspondence:
pierce.ogden@gmail.com (PJO);
gchurch@genetics.med.harvard.
edu (GMC)

Present address: [†]Manifold
Biotechnologies, Boston, United
States

Competing interest: See page
17

Reviewing Editor: Wenfeng
Qian, Chinese Academy of
Sciences, China

## eLife assessment

This study presents a **valuable** and comprehensive mutagenesis map of the AAV2 *rep* gene, which will undoubtedly capture the interest of scientists working with adeno-associated viruses and those engaged in the field of gene therapy. The thorough characterization of massive *rep* variants across multiple AAV production systems bolsters the claims made in the study, highlighting its utility in enhancing our understanding of Rep protein function and advancing gene therapy applications. The evidence presented is **convincing** and establishes a strong foundation that will stimulate and inform future research in the field.

## Introduction

Recombinant adeno-associated viruses (rAAVs) are a popular tool for the delivery of gene therapies as wild-type AAV is not pathogenic. Wild-type AAV consists of a 4.7 kb single-stranded DNA genome, which is packaged into an approximately 26 nm icosahedral capsid (*Srivastava et al., 1983*; *Xie et al., 2002*). The AAV genome is flanked on either end by 145-nucleotide inverted terminal repeats, which form hairpins, serve as the origins of viral replication, and are the only sequences required in *cis* for packaging of DNA into the capsid (*Xiao et al., 1997*). As such, the remainder of the AAV genome can be replaced with a gene of interest to generate rAAV vectors (*Samulski and Muzyczka, 2014*). The wild-type AAV genome consists of two genes, *rep* and *cap* (*Srivastava et al., 1983*). During rAAV production, these genes are supplied in *trans* to the inverted terminal repeat plasmid. The *cap* gene encodes three structural proteins, VP1, VP2, and VP3, which assemble in an approximately 1:1:10 ratio to form the 60-mer capsid (*Cassinotti et al., 1988*; *Wörner et al., 2021*). Engineering of the

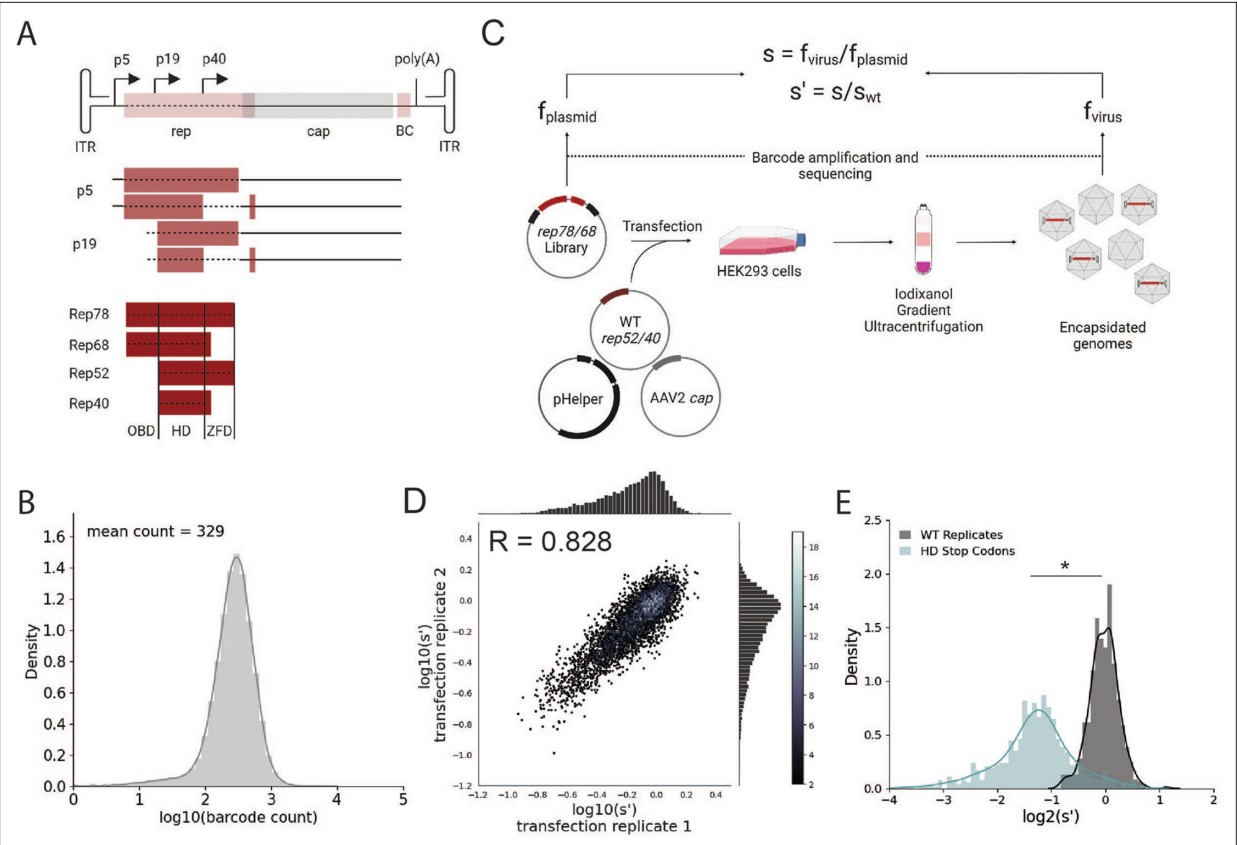

**Figure 1.** Comprehensive mutagenesis library design and production assay. (**A**) Organization of the AAV2 genome and Rep protein domains. Top: single-stranded DNA genome, middle: RNA transcripts, bottom: Rep proteins. Dotted lines indicate mutated regions. (**B**) Density plot of barcode counts in the pCMV-Rep78/68 plasmid library. (**C**) Overview of production assay for the pCMV-Rep78/68 library and calculation of wild-type normalized production fitness values (s'). (**D**) Amino acid level production fitness values from replicate transfections of the pCMV-Rep78/68 library. Pearson *R* correlation coefficient calculated after log transformation. (**E**) Density plot of production fitness values for wild-type (black) and premature stop codon (blue) controls for the pCMV-Rep78/68 library. *p<10$^{-20}$ (Mann–Whitney *U*-test). Panels (**A**) and (**C**) created with BioRender.com and published using a CC-BY-NC-ND license with permission.

The online version of this article includes the following figure supplement(s) for figure 1:

**Figure supplement 1.** Comprehensive mutagenesis library design and production assay results for WT AAV2 format library.

*cap* gene has enabled targeting of AAV vectors to specific tissues and cell populations. The *rep* gene encodes four proteins, Rep78, Rep68, Rep52, and Rep40, which are generated through the use of two promoters, p5 and p19, and alternative splicing (*Davis et al., 2000*).

The larger Rep proteins, Rep78 and Rep68, are required for genome replication while the smaller Rep proteins, Rep52 and Rep40, facilitate genome packaging. Notably, Rep78 and Rep68 alone are each sufficient for rAAV vector production (*Hölscher et al., 1995*). However, the presence of Rep52/40 enhances genome packaging and therefore rAAV titer (*Chejanovsky and Carter, 1989*; *King et al., 2001*). The *rep* gene encodes three protein domains: an origin-binding domain, a helicase domain, and a zinc-finger domain (*Figure 1A*; *Di Pasquale and Stacey, 1998*; *Im and Muzyczka, 1990*; *Smith and Kotin, 1998*). All four Rep proteins contain the helicase domain as well as a nuclear localization signal (*Cassell and Weitzman, 2004*). Only Rep78 and Rep68 contain the origin-binding domain and only Rep78 and Rep52 contain the zinc-finger domain. Additionally, residues in the linker domain between the origin-binding and helicase domains cooperate with residues at the N-terminus of the helicase domain to facilitate oligomerization of the larger Rep proteins, which is required for AAV production (*Zarate-Perez et al., 2012*).

The origin-binding domain contains three separate DNA-binding motifs that are important for genome replication (*Hickman et al., 2002*; *Hickman et al., 2004*). Firstly, the origin-binding domain recognizes the Rep-binding site, which consists of GCTC repeats present in the double-stranded region of the inverted terminal repeats (*Musayev et al., 2015a*). Unwinding of this double-stranded DNA by the helicase domain enables the origin-binding domain to interact with single-stranded DNA at its second motif, the active site pocket. Here, the origin-binding domain nicks the single-stranded DNA at the terminal resolution site, an essential step in viral genome replication (*Im and Muzyczka, 1990*; *Snyder et al., 1990*). Finally, the origin-binding domain contains a single-stranded DNA hairpin-binding site, which interacts with one of the inverted terminal repeat hairpins. Hairpin binding, however, is not required for terminal resolution site nicking or genome replication (*Wu et al., 1999*). The origin-binding domain can also recognize GCTC repeats present in the p5 promoter; in the presence of a helper virus, such as adenovirus, the Rep proteins activate transcription from the endogenous promoters and in the absence of helper virus the Rep proteins repress transcription (*Labow et al., 1986*; *Murphy et al., 2007*).

The helicase domain plays a definitive role in all steps of AAV production while the zinc-finger domain is likely dispensable for production. The helicase domain unwinds the double-stranded inverted terminal repeats during genome replication and its activity has been shown to facilitate genome packaging (*Brister and Muzyczka, 1999*; *King et al., 2001*). There is also evidence that residues within the helicase domain mediate capsid interactions. Mutations to the helicase domain of Rep52/40, as well as mutations near the fivefold axis of symmetry in the capsid, reduced the interaction between the Rep proteins and capsid and resulted in lower rAAV titers (*Bleker et al., 2006*; *King et al., 2001*). The third Rep domain, the zinc-finger domain, has not been shown to bind DNA but is involved in binding to various host cell proteins (*Di Pasquale and Stacey, 1998*). Previous work indicated that premature stop codons introduced into the zinc-finger domain-encoding region of *rep* do not affect rAAV titers, suggesting that this domain is not required for rAAV production (*Mietzsch et al., 2021*).

As more rAAV vectored therapies are approved, manufacturing sufficient rAAV quantities to meet patient need is an increasingly important issue. Engineering the Rep proteins is a promising avenue for improving AAV production. The Rep proteins are involved in all steps of viral production, including regulation of VP expression, genome replication, and genome packaging. However, the Rep proteins are not a structural component of the final vector. As such, the Rep proteins can be designed to optimize viral production without affecting downstream processes, such as cell targeting, which are driven by the capsid. The sequence of the *rep* genes in naturally occurring serotypes is relatively well conserved (average of 78% amino acid sequence identity between AAV2 *rep* and AAV1–13 *rep*). While previous mutational studies of the Rep proteins have been conducted, these studies have focused primarily on identifying non-functional Rep variants to identify the location of key motifs (*Davis et al., 2000*; *Gavin et al., 1999*; *Hörer et al., 1995*; *Yang et al., 1992*). As such, Rep engineering efforts may benefit from the exploration of additional sequence diversity.

Toward this end, we generated a library of all possible single-codon mutations of the AAV2 *rep* gene, the *rep* gene most commonly used for AAV production. We assayed the effect of these mutations on AAV2 production and generated a detailed sequence-to-function map of the AAV2 *rep* gene.

## Results

### Comprehensive mutagenesis assays the effect of all single-codon mutations in the AAV2 *rep* gene on AAV2 production

To better understand how mutations to *rep* affect AAV production, we generated two *rep* libraries. The first library, referred to as pCMV-Rep78/68, consisted of a cytomegalovirus (CMV) promoter followed by the *rep* open-reading frame with an M225G mutation introduced to prevent expression of the smaller Rep proteins. This library allowed us to assay the effect of mutating only Rep78 and Rep68. The second library, referred to as WT AAV2, was analogous to the sequence of the wild-type virus and contained the p5 promoter and *rep* and *cap* open-reading frames. Use of the endogenous promoters in the WT AAV2 library allowed us to capture the effect of *rep* mutations on Rep and VP expression. This library also enabled simultaneous mutation of all four Rep proteins. For each library, we generated all 39,297 possible single-codon substitution and deletion variants spanning from the

**Table 1.** Percent of expected variants sequenced in plasmid and viral libraries.

| | pCMV-Rep78/68 | | WT AAV2 | |
| --- | --- | --- | --- | --- |
| | Plasmid (%) | Viral (%) | Plasmid (%) | Viral (%) |
| Barcodes | 99.8 | 98.0 | 98.3 | 97.6 |
| Codon variants | 100 | 99.9 | 99.7 | 99.7 |
| Amino acid variants | 100 | 100 | 99.9 | 99.9 |

Rep78/68 start codon to the Rep78/52 stop codon (*Figure 1A*). All variants contained unique 20 bp barcodes at the 3′ end of the *rep* gene (pCMV-Rep78/68 library) or *cap* gene (WT AAV2 library), and each codon variant was represented by a minimum of two barcode sequences. To assess the diversity of the plasmid libraries, we amplified and sequenced the barcodes from each of the plasmid pools (*Figure 1B*, *Figure 1—figure supplement 1A*, and *Table 1*). We observed 100 and 99.9% of the expected amino acid variants in the pCMV-Rep78/68 and WT AAV2 libraries, respectively.

Next, we transfected the plasmid libraries into HEK293T cells to assay the effect of all single-codon mutations on the production of genome-containing viral particles (*Figure 1C*). Production fitness values for each variant were calculated and normalized to internal wild-type controls as shown in *Figure 1C*. For each library, we performed transfections in duplicate and compared the production fitness values calculated from each replicate (*Figure 1D*, *Figure 1—figure supplement 1B*). For both libraries, production fitness values were well correlated across replicates, indicating that the genotype–phenotype linkage was maintained during viral production. After examining the correlation between biological replicates, we next looked at the distribution of fitness values for the wild-type and premature stop codon controls (*Figure 1E*, *Figure 1—figure supplement 1C*). As expected, premature stop codons had a deleterious effect on AAV2 production and fitness values for replicate wild-type controls clustered together.

## Annotation of the AAV2 *rep* sequence-to-function map

To better understand the sequence-to-function map of the AAV2 Rep proteins, we visualized the data from our production assay in multiple ways. First, we generated heatmaps containing the wild-type normalized production fitness values for all single amino acid substitutions and deletions (*Figure 2*, *Figure 2—figure supplement 1*). Additionally, we calculated the 'mutability' of each residue by averaging the normalized production fitness values at each position across all substitutions. We mapped the mutability of each residue onto structures of the origin-binding domain in complex with the Rep-binding site (PDB: 4ZQ9, *Figure 3A and B*), the origin-binding domain in complex with single-stranded DNA from the inverted terminal repeat hairpin (PDB: 6XB8, *Figure 3C*), the origin-binding domain alone (PDB: 5D6X, *Figure 3—figure supplement 1A*), and the helicase domain (PDB: 1S9H, *Figure 3—figure supplement 1B*; *James et al., 2003*; *Musayev et al., 2015a*; *Musayev et al., 2015b*; *Santosh et al., 2020*). Finally, we used the production fitness values calculated for individual barcodes to determine which amino acid changes resulted in production fitness values significantly different from wild-type (*Figure 2—figure supplement 3A and B*). We observed that the origin-binding and zinc-finger domains were more tolerant of mutation than the helicase domain. Interestingly, a stark boundary between amino acids D212 and A213 was observed; at residue A213, the larger Rep proteins become much less tolerant of mutation. This boundary is adjacent to linker domain residues P214-Y224, which were previously reported to be important for Rep78/68 oligomerization (*Musayev et al., 2015a*; *Zarate-Perez et al., 2012*).

When looking across all library members, the production fitness values determined for the pCMV-Rep78/68 and WT AAV2 libraries were well correlated (*Figure 2—figure supplement 2A*). However, as expected, introduction of a methionine upstream of the Rep52/40 start codon was deleterious in the WT AAV2 format but not in the pCMV-Rep78/68 format, where Rep52/40 were expressed in *trans*.

The majority of beneficial substitutions clustered in the origin-binding domain. In particular, substitutions between residues V11 and D62 of the origin-binding domain enhanced AAV2 production relative to wild-type. These residues are involved in recognition of the inverted terminal repeat hairpin (*Figure 3C*; *Musayev et al., 2015a*). Origin-binding domain–hairpin interactions have been shown to

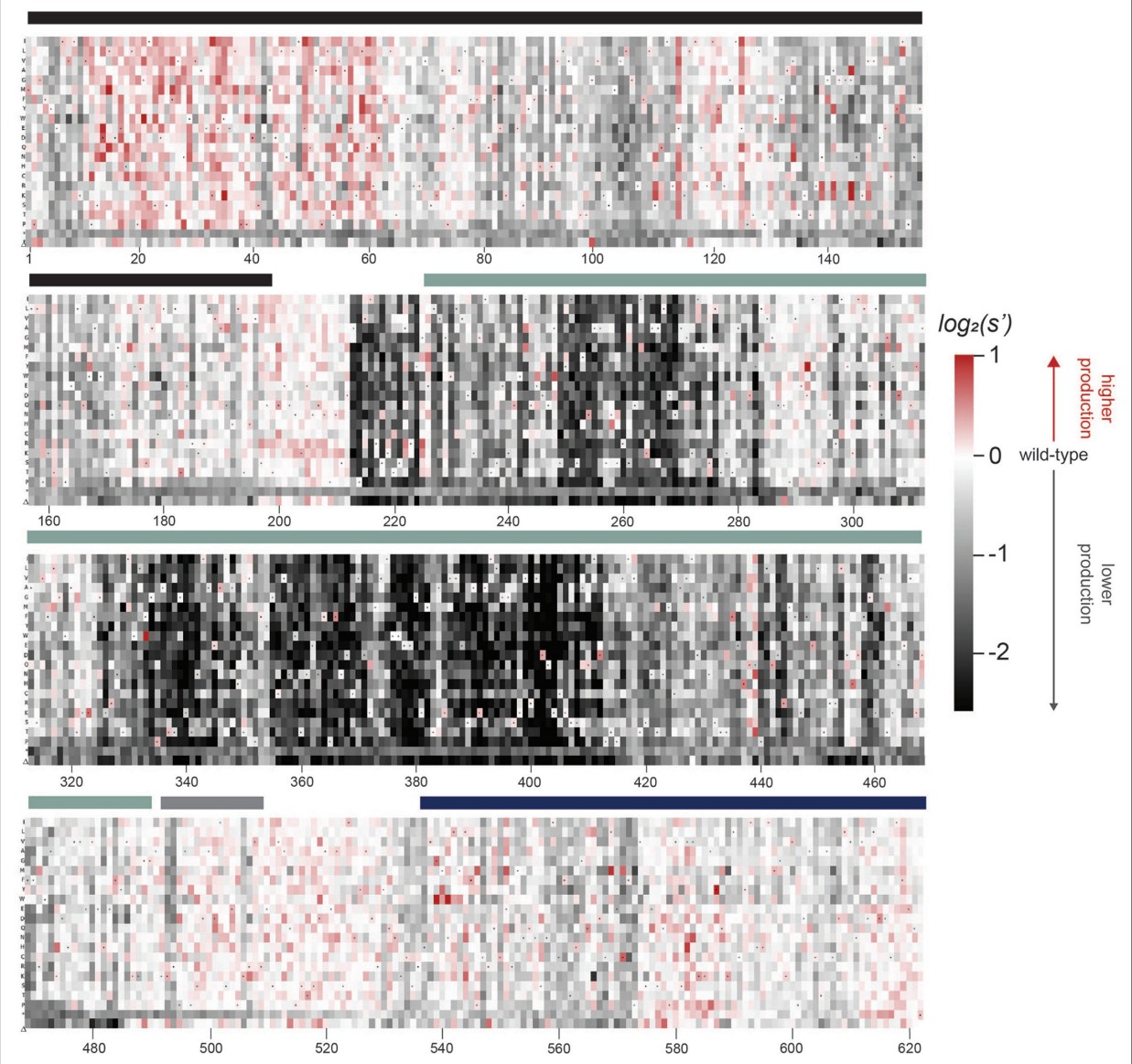

**Figure 2.** Effects of all single amino acid substitutions and deletions in the Rep78/68 proteins on AAV2 production. Amino acid level production fitness values from the pCMV-Rep78/68 production assay were calculated as in *Figure 1C* by summing barcode counts for synonymous mutations. Rectangles are colored by mutational effect on the production of genome-containing particles, with black indicating deleterious mutations and red indicating beneficial mutations. Colored bars above the heatmaps indicate protein domains. Black: origin-binding domain; light blue: helicase domain; gray: nuclear localization signal; and navy blue: zinc-finger domain. Black dots indicate wild-type amino acid identity.

The online version of this article includes the following figure supplement(s) for figure 2:

**Figure supplement 1.** Effects of all single amino acid substitutions and deletions in Rep78, Rep68, Rep52, and Rep40 on AAV2 production.

**Figure supplement 2.** Mutations to AAV2 *rep* have similar effects on AAV2 production in pCMV-Rep78/68 and WT AAV2 format libraries.

**Figure supplement 3.** Comparison of Rep variant and wild-type production fitness values.

**Figure supplement 4.** Codon level production fitness values for the pCMV-Rep78/68 format library.

**Figure supplement 5.** Codon level production fitness values for the WT AAV2 format library.

**Figure supplement 6.** The distribution of production fitness values is narrower for synonymous variants than for nonsynonymous variants.

**Figure supplement 7.** Inclusion of synonymous codon variants enables interrogation of nucleotide-level effects.

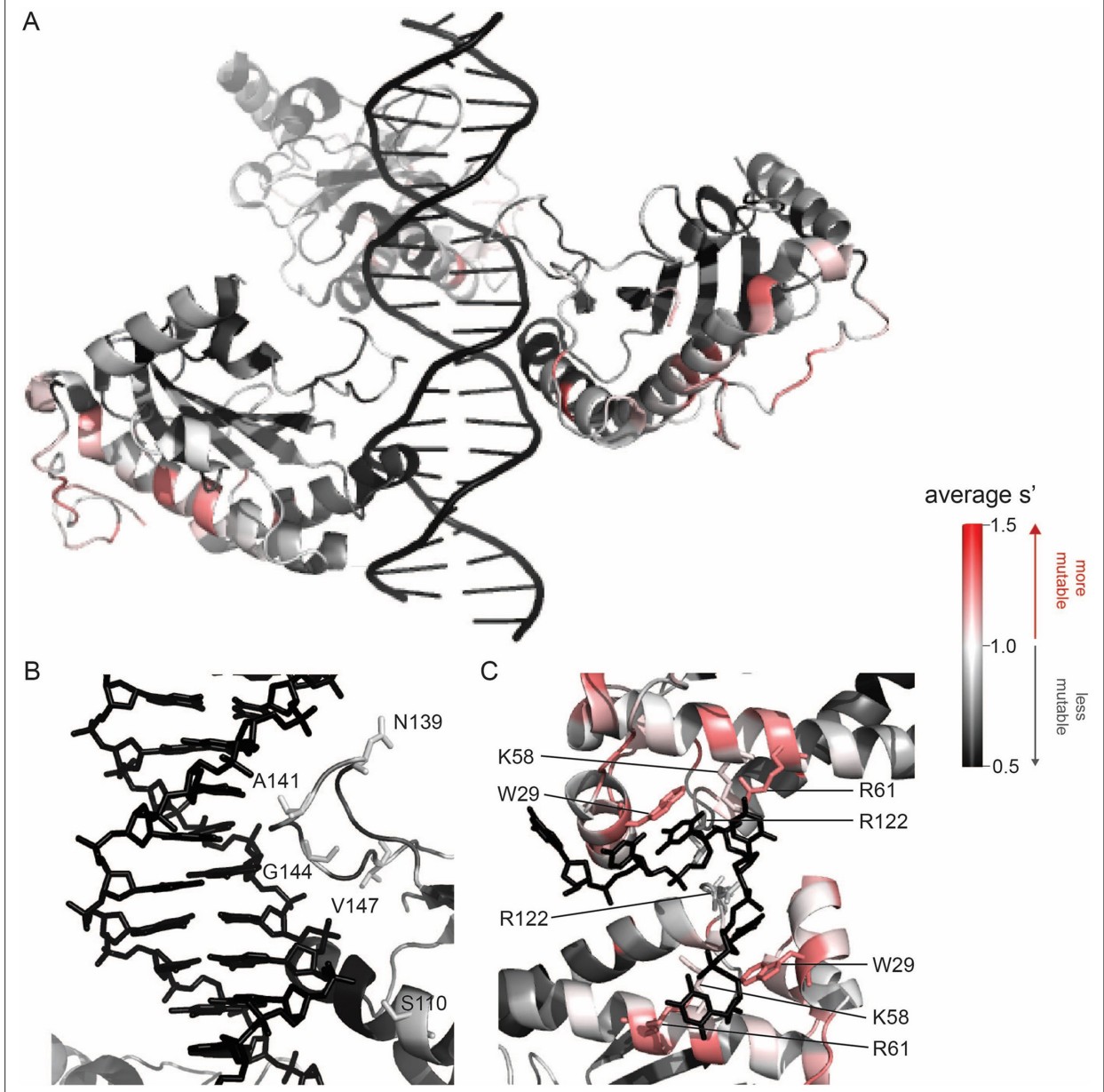

**Figure 3.** Beneficial substitutions cluster in DNA-interacting residues. (**A**) Average production fitness values for all substitutions at each position mapped onto the structure of the origin-binding domain in complex with the AAVS1 Rep-binding site (PDB 4ZQ9). (**B**) Closeup view of origin-binding domain–Rep-binding site interactions. Residues where substitutions to positively charged residues are beneficial are shown as sticks. (**C**) Average production fitness values mapped onto the structure of the origin-binding domain in complex with the single-stranded inverted terminal repeat hairpin (PDB 6XB8). DNA-interacting residues are shown as sticks. Residues are colored by mutability, with red indicating higher mutability and black indicating lower mutability.

The online version of this article includes the following figure supplement(s) for figure 3:

**Figure supplement 1.** Average production fitness values from the pCMV-Rep78/68 library production assay mapped onto (**A**) the structure of the origin-binding domain active site (PDB 5DCX) and (**B**) the structure of the helicase domain (PDB 1S9H).

enhance terminal resolution site nicking but are not required for nicking to occur (***Wu et al., 1999***). The majority of Rep-binding site-interacting residues, on the other hand, were intolerant of mutation (***Figure 3B***). The origin-binding domain–Rep-binding site interaction is important in mediating Rep–genome interactions during both genome replication and promoter regulation (***Labow et al., 1986***; ***Murphy et al., 2007***; ***Musayev et al., 2015b***). Interestingly, there are a handful of Rep-binding site-interacting residues that were somewhat mutable, including S110 and N139, which form contacts

with the phosphate backbone, and A141, which forms contacts with bases in the Rep-binding site. Mutation of S110, N139, and A141, as well as G144 and V147, to positively charged residues had a beneficial effect on AAV2 production. Residues N139, A141, G144, and V147 are part of the loop that connects β-strand 4 to α-helix E (*Musayev et al., 2015b*). The sequence of this loop is highly variable across serotypes (*Musayev et al., 2015a*). Notably, the asparagine at position 139 is positively charged in AAV5 (K139). However, no serotypes contain positively charged amino acids at positions 141, 144, or 147. Our data indicate that substitution of DNA-interacting residues in the origin-binding domain can enhance AAV production and identifies beneficial substitutions not observed in nature.

A clear pattern in the location of mutable residues in the origin-binding domain active site can also be observed (*Figure 3—figure supplement 1A*). This active site is responsible for cleaving single-stranded DNA and is formed by the origin-binding domain beta sheet and Y156, the active site nucleophile (*Hickman et al., 2004*). Residues with side chains directed toward the active site are less mutable than adjacent residues with their side chains directed away from the active site. It is more difficult to discern positional trends in mutability within the helicase domain as it is less tolerant of mutation than the origin-binding domain (*Figure 3—figure supplement 1B*). No structures of the ZFD are available. However, the majority of variants in this domain do not have production fitness values that are significantly different from wild-type (*Figure 2—figure supplement 3A and B*).

## Generation of all single-codon variants enables interrogation of nucleotide-level effects

As our libraries contain all possible single-codon mutations, we were able to search for variants with nucleotide-level effects on production by comparing differences in production fitness values between synonymous codon variants (*Figure 2—figure supplements 4 and 5*). In general, there was good agreement in the fitness values for synonymous variants. The production fitness values for synonymous codon variants were more tightly distributed than the fitness values for nonsynonymous variants (*Figure 2—figure supplement 6A and B*). Comparison of fitness values between synonymous codons that do and do not introduce premature stop codons into alternate reading frames allowed us to search for possible frameshifted open-reading frames. However, no evidence of frameshifted open-reading frames was observed (*Figure 2—figure supplement 7A and B*).

We did observe an interesting pattern at amino acid Y283 (*rep* nucleotides 847–849). Variants with a c.849C>G mutation had lower production fitness values than synonymous variants without a c.849C>G (p<10⁻⁸, Mann–Whitney *U*-test). The deleterious effect of this mutation can also be observed by plotting the average fitness value at each nucleotide position for each of the four bases (*Figure 2—figure supplement 7C and D*). The negative effect of a c.849C>G mutation was consistent across the pCMV-Rep78/68 and WT AAV2 library formats and was also observed in experiments with alternate *cap* genes. In each library, there were 15 codon variants with a c.849C>G mutation, 13 of which had synonymous codons without a c.849C>G mutation for comparison. Our results indicate

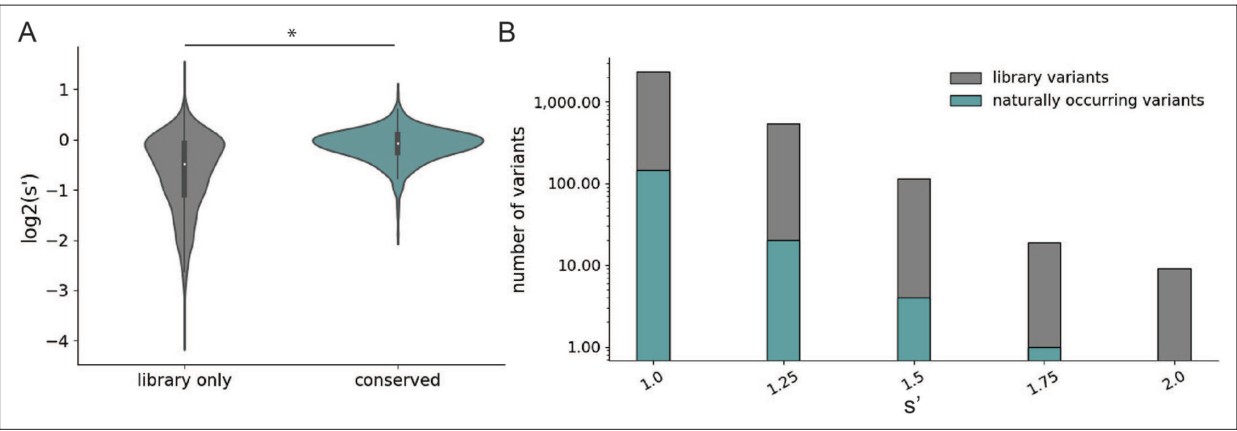

**Figure 4.** Comparison of comprehensive mutagenesis measurements to variation in nature. (**A**) Distribution of wild-type normalized production fitness values for conserved variants (blue) and variants found only in the library (gray). (**B**) Total number of variants and number of conserved variants with s' greater than wild-type (s' > 1). *p<10⁻²⁰ (Mann–Whitney *U*-test).

that the nucleotide sequence of the *rep* gene is optimized. However, we were not able to detect broad nucleotide-level effects in our data.

## Comprehensive mutagenesis identifies beneficial variants not observed in nature

We observed that mutations to amino acids found in other naturally occurring serotypes were better tolerated than mutations to amino acids that are not found in nature (*Figure 4A*). However, a majority of the variants with production fitness values greater than that of wild-type are not observed in nature (*Figure 4B*). Only 245/2351 (6.17%) of variants with s' >1 and only 4/115 (3.48%) of variants with s' > 1.5 are observed in AAV serotypes 1–13. These data emphasize the power of our comprehensive approach to identify novel functional sequence diversity.

## Validation of multiplexed production assay results

To validate the results of our multiplexed production assay, we selected 14 variants, cloned them individually into the pCMV-Rep78/68 format, and determined their effect on production by measuring DNase-resistant particle titers (*Figure 5A*). We included H92A and K340H, mutations to the origin-binding domain and helicase domain active sites, respectively, as controls (*Musayev et al., 2015a*; *Smith and Kotin, 1998*). Twelve variants with mutations in the origin-binding domain, linker domain, and helicase domain and fitness values greater than wild-type were also assayed. Finally, we included two deleterious variants identified in our multiplexed assay, N139G and A213V. The DNase-resistant particle titers determined from individual transfections are well correlated with the fitness values determined from the multiplexed production assay (*Figure 5B*). We performed similar validation for a panel of WT AAV2 format variants (*Figure 2—figure supplement 2B*).

To enable multiplexed analysis of *rep* variants in our library assay, the mutant *rep* genes themselves are packaged into the AAV capsids. Relatively small amounts of *rep* library plasmids are also used in transfection to reduce the frequency at which multiple *rep* variant plasmids enter the same cell. These procedures maintain a genotype–phenotype linkage and allow us to identify the *rep* sequences that enable production of genome-containing particles. We sought to confirm that the effects observed in the library production assay would be conserved when the *rep* variants were supplied in *trans* to the inverted terminal repeat genome, as in the traditional triple-plasmid transfection method used for rAAV production. To this end, we selected a small subset of *rep* variants and cloned them into an AAV2 pRepCap plasmid without inverted terminal repeats. Here, we assayed variants with single-codon mutations in each of the Rep domains.

We used these mutant pRepCap plasmids to produce rAAV vectors containing four different genomes, three single-stranded genomes and one self-complementary genome. The DNase-resistant particle titers for these pRepCap variants were relatively consistent across the different genomes (*Figure 5C*). Importantly, the DNase-resistant particle titers determined when the *rep* variants were supplied in *trans* correlated well with the normalized fitness values determined in our production assay (*Figure 5D*). Several single-codon variants, including S110R, N139R, and K566L, showed 50–100% improvements in DNase-resistant particle titer over wild-type. Western blot analysis indicated that most of these variants had little to no effect on Rep protein or VP expression (*Figure 5E*). Interestingly, in the case of the deleterious variants, A213V and K340H, Rep52/40 protein levels were increased. Codon GCG213 is located just upstream of the Rep52/40 start codon. As such, mutations to this position likely affect the strength of the p19 promoter and therefore Rep52/40 expression levels. Notably, the effects of this mutation on the p19 promoter are separate from the deleterious effects of the A213V amino acid change; the negative effects of the A213V change are observed even when Rep52/40 are expressed in *trans* (*Figure 5A*).

## The S110R variant increases transducing particle titer

We determined the relative transduction efficiency of rAAV2 produced with wild-type Rep and two different Rep variants, S110R and Q439T. We set up individual transfections with these Rep variants and affinity purified the resulting rAAV2 vectors. Use of the S110R Rep variant resulted in approximately twofold higher viral genome titers (*Figure 5—figure supplement 1A*). We also performed capsid ELISAs and determined that the S110R variant resulted in a similar increase in total particle titers; while viral genome titers were increased relative to wild-type Rep, the ratio of viral genomes:

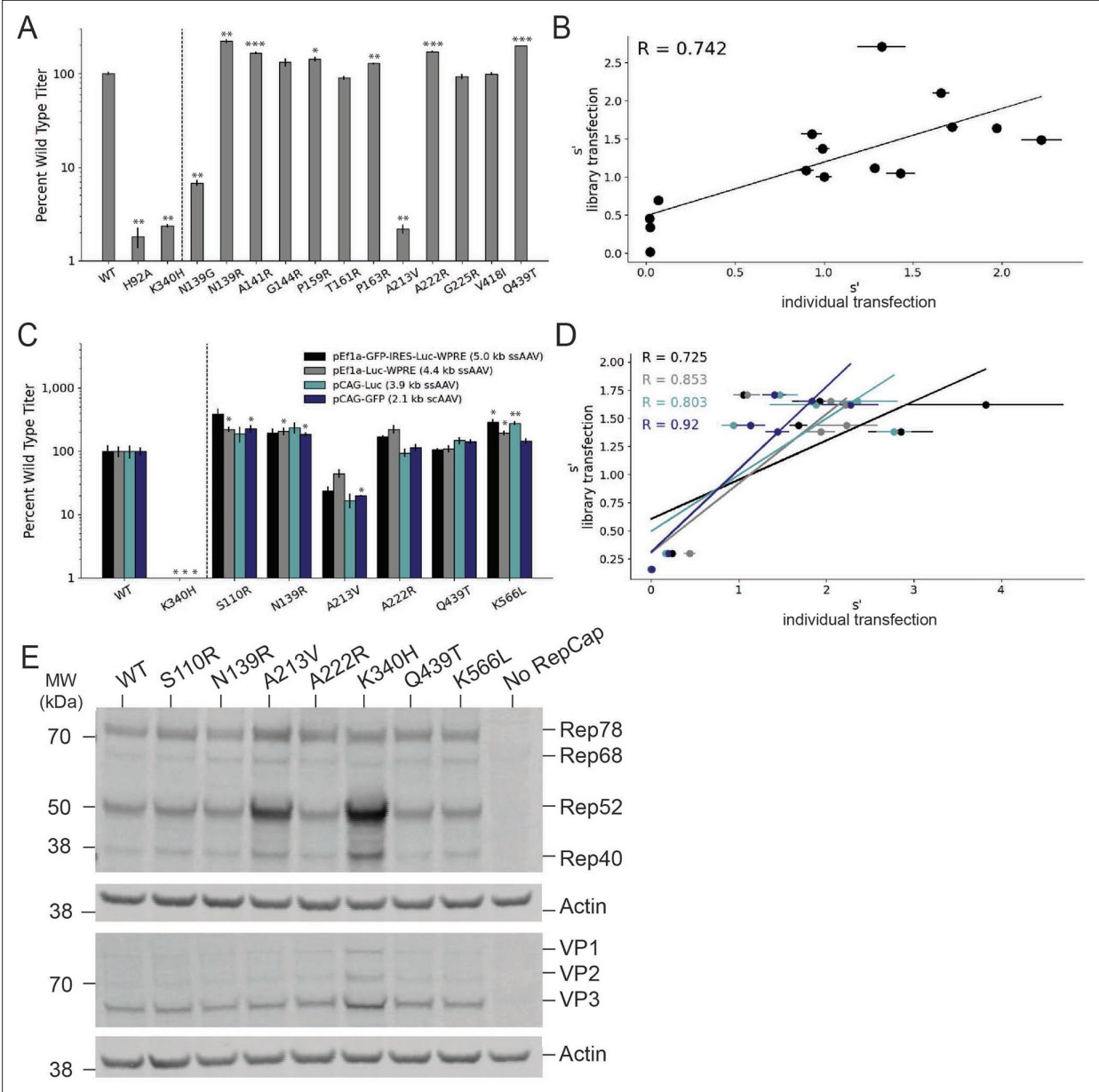

**Figure 5.** Validation of AAV2 library production assay results. (**A**) DNase-resistant particle titers for 14 single amino acid pCMV-Rep78/68-inverted terminal repeat variants produced individually. Titers for previously characterized variants are plotted to the left of the dotted line. (**B**) Relationship between normalized production fitness values from library experiments and DNase-resistant particle titers from individual transfections. (**C**) DNase-resistant particle titers for four recombinant adeno-associated virus (rAAV) genomes produced with the indicated Rep variants. Titers for previously characterized variants are plotted to the left of the dotted line. (**D**) Relationship between rAAV DNase-resistant particle titers and normalized production fitness values from library experiments. (**E**) Expression of Rep and VP proteins from variant pRepCap plasmids by western blot. For panels (**A**) and (**C**), asterisks indicate significance of titer differences between the Rep variant and the relevant wild-type control. *p<0.05, **p<0.01, ***p<0.001 (Welch's t-test).

The online version of this article includes the following figure supplement(s) for figure 5:

**Figure supplement 1.** Effect of single amino acid Rep substitutions on the viral genome titer, physical particle titer, and relative transduction efficiency of affinity purified rAAV2.

**Table 2.** Viral genome and physical particle titers for rAAV2 produced with Rep variants.

| Rep variant | Replicate | qPCR titer (vg/mL) | qPCR SD (vg/mL) | ELISA titer (capsids/mL) | ELISA titer SD (capsids/mL) | Viral genome:capsid titer ratio |
|---|---|---|---|---|---|---|
| WT | A | 6.18E+11 | 5.77E+10 | 7.82E+11 | 1.04E+11 | 0.79 |
| | B | 3.00E+11 | 3.85E+10 | 1.77E+11 | 8.95E+10 | 1.70 |
| Q439T | A | 7.01E+11 | 6.47E+10 | 8.14E+11 | 4.27E+10 | 0.86 |
| | B | 4.55E+11 | 2.33E+10 | 3.61E+11 | 1.21E+11 | 1.26 |
| S110R | A | 1.45E+12 | 3.24E+10 | 1.73E+12 | 1.56E+11 | 0.84 |
| | B | 1.04E+12 | 6.81E+10 | 1.43E+12 | 1.31E+11 | 0.73 |

capsids was unchanged (*Table 2*). We then used a constant volume of each rAAV2 prep to transduce HEK293T cells and quantified the relative amount of transducing particles by measuring transgene expression (luciferase activity, *Figure 5—figure supplement 1B*). Again, an approximately twofold increase in relative transduction was observed with rAAV2 produced with S110R Rep as compared to wild-type Rep.

## Mutations in the AAV2 *rep* gene have similar effects on the production of AAV2, AAV5, and AAV9 capsids

Many different AAV capsid serotypes are of clinical interest for their unique tissue-targeting properties. Given the physical interaction between Rep proteins and the capsid, we hypothesized that some *rep* mutations would have unique effects on the production of different capsid serotypes. To identify *rep* variants with serotype-specific effects on production, we repeated our pCMV-Rep78/68 library production assay using AAV5 and AAV9 *cap* genes in place of AAV2 (*Figure 6—figure supplements*

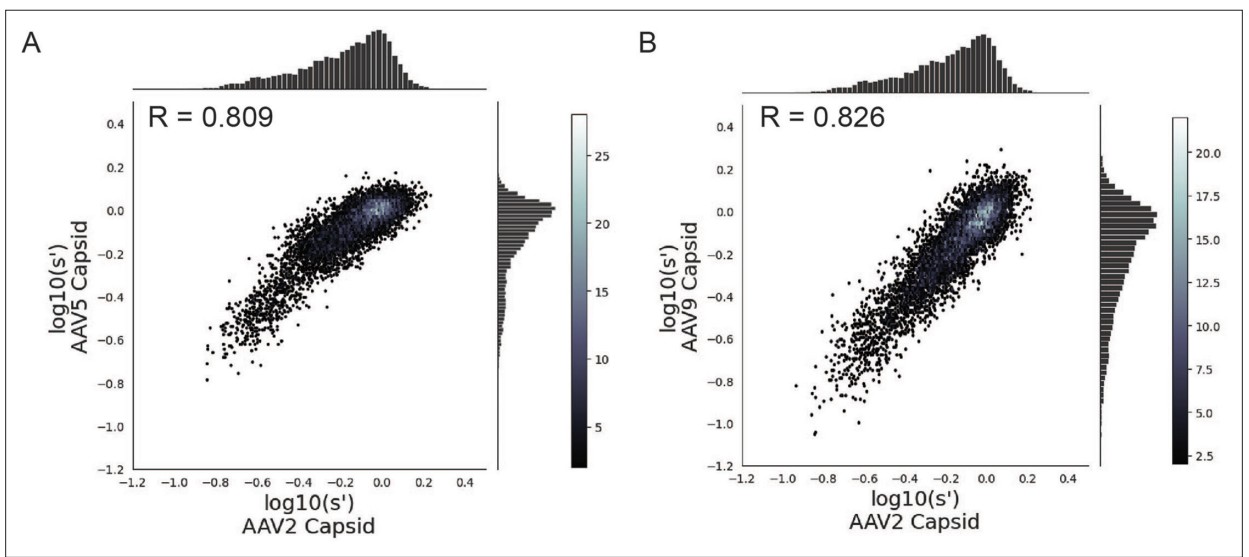

**Figure 6.** Mutations to AAV2 rep have similar effects on AAV2, AAV5, and AAV9 capsid production. Wild-type normalized production fitness values from the library production assay with (**A**) AAV5 and AAV2 capsids and (**B**) AAV9 and AAV2 capsids. Pearson *R* correlation coefficient calculated after log transformation.

The online version of this article includes the following figure supplement(s) for figure 6:

**Figure supplement 1.** Effects of all single amino acid substitutions and deletions in the Rep78/68 proteins on AAV5 capsid production.

**Figure supplement 2.** Effects of all single amino acid substitutions and deletions in the Rep78/68 proteins on AAV9 capsid production.

**Figure supplement 3.** Codon level production fitness values for the AAV5 capsid production assay.

**Figure supplement 4.** Codon level production fitness values for the AAV9 capsid production assay.

**Figure supplement 5.** DNase-resistant particle titers for AAV2, AAV5, and AAV9 capsids produced individually with the indicated Rep variants.

*1–4*). The AAV5 capsid was chosen as it has low overall sequence identity with the AAV2 capsid (57.7% amino acid identity). The AAV9 capsid is more similar to the AAV2 capsid (81.7% amino acid identity) but is of particular clinical interest and differs from AAV2 at amino acid positions 329 and 330, which were previously identified as important for Rep–capsid interactions (AAV2: T329/T330; AAV9: V331/K332) (*Bleker et al., 2006*). Surprisingly, the correlation in fitness values across capsid serotypes was comparable to the correlation between biological replicates, indicating that the effect of AAV2 *rep* variants on production is largely consistent across capsid serotypes (*Figure 6A and B*). We individually produced rAAV5 and rAAV9 vectors using a subset of our *rep* variants and confirmed that the titers with each *rep* variant are well correlated across capsid serotype (*Figure 6—figure supplement 5*).

## Discussion

We have generated a comprehensive mutagenesis library of the AAV2 *rep* gene, containing all possible single-codon substitutions and deletion variants. Multiplexed assay of this library allowed us to generate a sequence-to-function map, linking all variants to their effect on AAV production.

Many of the previous AAV mutagenesis studies have focused on the AAV *cap* gene. Previous work in our lab generated a library containing all possible single-codon substitutions, deletions, and insertions of the AAV2 *cap* gene and assayed their effect on AAV2 production (*Ogden et al., 2019*). Notably, we observed a smaller range of fitness values in our production assay with the *rep* library than with the previously reported *cap* library. This observation aligns with our knowledge of Rep and capsid biology. The Rep proteins' primary function is to facilitate AAV production, while the capsid has evolved to not only enable assembly and genome packaging, but also to facilitate cell targeting, entry, and nuclear trafficking. Additionally, the Rep proteins possess multiple enzymatic activities that require the proteins to adopt specific and dynamic conformations. The capsid, in contrast, is not known to possess enzymatic activity outside of the phospholipase domain located at the N-terminus of VP1 (*Stahnke et al., 2011*). It follows that there are greater mutational constraints on *rep* than on *cap* in the context of AAV2 production.

Despite the smaller range of fitness values observed here, we identified numerous Rep variants with production fitness values greater than or equal to that of wild-type AAV2 Rep that are not observed in naturally occurring AAV serotypes. We attribute this discrepancy to the fact that optimal rAAV production may not equate to optimal fitness of AAV in the endogenous host. Wild-type AAV has both a lytic and a latent cycle (*Liu, 2014*). The latent cycle occurs when no helper virus is present. Rep and VP expression are repressed and the AAV genome is integrated into the host cell genome in a process that also involves Rep (*Pereira et al., 1997*; *Surosky et al., 1997*). In the presence of a helper virus, such as adenovirus, the lytic cycle proceeds. In this instance, Rep78/68 activate transcription from the AAV promoters and genome replication and packaging occur (*Pereira et al., 1997*). rAAV production is somewhat analogous to the lytic cycle. However, naturally occurring AAV must balance the effect of any mutations on fitness in both the lytic and latent contexts. An additional explanation for this finding is the relatively small number of AAV serotypes, for which rep sequences are available. In our analysis, we compared the beneficial variants identified in our library to the *rep* sequences from 12 different AAV serotypes (AAV1–13, no sequence for AAV9 *rep*). This is a small number of naturally occurring *rep* sequences when compared to the size of our library. While there have been efforts to identify additional naturally occurring *cap* sequences, there has been relatively little effort to expand the number of naturally occurring *rep* sequences (*Wang et al., 2019*). We expect that such work may identify *rep* sequences containing many of the beneficial variants identified in our library.

We observed that substitutions in the origin-binding and zinc-finger domains were better tolerated than substitutions in the linker and helicase domains. A sharp drop in mutability was observed between residues D212 and A213. Previous work has identified residues V215 through Y224 as the minimal linker sequence required to facilitate Rep78/68 oligomerization (*Zarate-Perez et al., 2012*). In a separate study, it was reported that mutation of P214 reduced the ability of Rep to oligomerize (*Musayev et al., 2015a*). Our data indicate that residue A213 is intolerant of mutation. A213 likely plays an important role in AAV production and, given its position, may be involved in Rep78/68 oligomerization. Additional work is needed to confirm that mutations to A213 affect AAV production by interfering with Rep78/68 oligomerization. In both library formats, the zinc-finger domain was relatively tolerant of mutation and few substitutions in this domain resulted in production fitness values that were significantly different from wild-type. At some positions, even

introduction of a premature stop codon was not deleterious. It was recently reported that premature stop codons introduced at positions 522 and 553 of Rep did not affect AAV2 production when supplied in a pRepCap plasmid (*Mietzsch et al., 2021*). Our results provide further evidence that the zinc-finger domain is dispensable for AAV production and identify A213 as a potential linker domain residue.

The majority of beneficial substitutions clustered in the origin-binding domain. In a recent investigation of *rep* hybrids, *Mietzsch et al., 2021* reported that replacement of the entire AAV2 origin-binding domain with that of AAV1 or AAV8 improved the proportion of full capsids. Our data supports the importance of the origin-binding domain for AAV production and identifies specific origin-binding domain regions where beneficial mutations cluster. Most substitutions of inverted terminal repeat-hairpin-interacting residues and substitutions of residues in the Rep-binding site-interacting loop to positively charged amino acids were beneficial. Previous studies have demonstrated that Rep binding to the inverted terminal repeat-hairpin improves the efficiency of terminal resolution site nicking but is not required for the nicking reaction to occur (*Wu et al., 1999*). The Rep-binding site-interacting loop, on the other hand, is important for the recognition of double-stranded DNA during genome replication and promoter binding. Interestingly, further characterization of the S110R variant, which falls in the Rep-binding site-interacting region, indicated that it increases viral genome and transducing particle titer relative to wild-type Rep but did not increase the ratio of viral genome: capsids, a proxy for the proportion of full capsids. Additional work is needed to understand which, if any, of the beneficial OBD variants identified here also improve the proportion of full capsids. Taken together, our results indicate that enhancement of Rep–DNA interactions may be a fruitful avenue for further improvement of AAV production.

Inclusion of all single-codon variants in our libraries allowed us to investigate the effect of synonymous mutations on AAV production. Interestingly, we observed that introduction of a c.849C>G mutation resulted in lower production fitness values compared to synonymous variants. Interestingly, this position is located downstream of the p5 and p19 promoters. Additionally, this mutation had a negative effect on production even when the *cap* gene was supplied in *trans*, making the activity of the p40 promoter irrelevant for AAV production. The c.849C>G mutation also does not fall near any known AAV2 splice sites (*Stutika et al., 2016*). While the mechanism by which this mutation affects production remains unclear, our results emphasize that the *rep* gene is optimized for AAV production at both the amino acid and nucleotide levels.

While we were able to discern a trend in the effect of synonymous variants at nucleotide position 849 of *rep*, the selection values for synonymous codon variants in general were relatively closely distributed. This precluded further investigation of the effect of synonymous codons on AAV production. Synonymous variants likely have an effect on aspects of AAV production, such as genome replication, transcriptional regulation, mRNA stability, and protein expression. However, our assay measures the aggregate effect of *rep* variants on all steps in the AAV production process and is likely unable to detect the effects of synonymous variants on specific steps in this process if those steps are not rate-limiting.

In an attempt to identify Rep substitutions with a beneficial effect on the production of specific capsid serotypes, we repeated the pCMV-Rep78/68 library assay using AAV5 and AAV9 *cap* genes in place of AAV2 *cap*. Surprisingly, the correlation in fitness values across experiments was similar to the correlation between biological replicates, suggesting that the majority of AAV2 *rep* mutations have a similar effect on the production of AAV2, AAV5, and AAV9 capsids. These results suggest that either *rep* mutations have a similar effect on Rep78/68–capsid interaction across serotypes, the effect of perturbing Rep78/68–capsid interactions is obscured by the deleterious effects of the same mutations on other Rep activities, or Rep78/68–capsid interactions are not a limiting factor in AAV production.

We have generated a comprehensive sequence-to-function map of the effect of all single-codon AAV2 *rep* mutations on AAV2, AAV5, and AAV9 production. Our experiments identified thousands of functional Rep variants, laying the groundwork for further engineering of these proteins and enhancement of large-scale gene therapy production.

## Materials and methods

### Key resources table

| Reagent type (species) or resource | Designation | Source or reference | Identifiers | Additional information |
|---|---|---|---|---|
| Cell line (*Homo sapiens*) | HEK293T | ATCC | ATCC:CRL-3216 | |
| Recombinant DNA reagent | pCMV-Rep78/68 plasmid library | This paper | RRID:Addgene_198050 | Complete list of plasmid sequences included in library can be found in the GitHub repo linked below |
| Recombinant DNA reagent | pEf1a-Luc-WPRE (ssAAV) plasmid | Addgene | RRID:Addgene_87951 | |
| Recombinant DNA reagent | pCAG-Luc (ssAAV) plasmid | Addgene | RRID:Addgene_83281 | |
| Recombinant DNA reagent | pCAG-GFP (scAAV) plasmid | Addgene | RRID:Addgene_83279 | |
| Sequence-based reagent | 5'-GAGCATCTGCCCGGCATTTC-3' | This paper | | Forward primer, binds to 5' region of AAV2 *rep* gene |
| Sequence-based reagent | 5'-ATCTGGCGGCAACTCCCATT-3' | This paper | | Reverse primer, binds to 5' region of AAV2 *rep* gene |
| Sequence-based reagent | 5'-HEX-ACAGCTTTG-ZEN-TGAACTGGGTGGCCGA-3IABkFQ-3' | This paper | | HEX-based probe, binds to 5' region of AAV2 *rep* gene |
| Sequence-based reagent | 5'-GAACCGCATCGAGCTGAA-3' | This paper | | Forward primer, binds to GFP sequence |
| Sequence-based reagent | 5'-TGCTTGTCGGCCATGATATAG-3' | This paper | | Reverse primer, binds to GFP sequence |
| Sequence-based reagent | 5'-FAM-ATCGACTTC-ZEN-AAGGAGGACGGCAAC-3IABKFQ-3' | This paper | | FAM-based probe, binds to GFP sequence |
| Sequence-based reagent | 5'-CATGTACCGCTTCGAGGAG-3' | This paper | | Forward primer, binds to luciferase sequence |
| Sequence-based reagent | 5'-GAAGCTAAATAGTGTGGGCACC3' | This paper | | Reverse primer, binds to luciferase sequence |
| Sequence-based reagent | 5'-FAM-CTTGCGCAG-ZEN-CTTGCAAGACTATAAGATTC-3IABKFQ-3' | This paper | | FAM-based probe, binds to luciferase sequence |
| Antibody | Anti-Rep 303.9 (mouse monoclonal) | American Research Products | American Research Products: 03-61069; RRID:AB_1540388 | 1:100 |
| Antibody | B1 anti-VP (mouse monoclonal) | American Research Products | American Research Products: 03-61058; RRID:AB_1540385 | 1:250 |
| Antibody | Anti-β-actin (rabbit polyclonal) | Abcam | Abcam: ab8227; RRID:AB_2305186 | 1:20,000 |
| Antibody | Anti-mouse IgG IRDye 800CW (goat monoclonal) | LI-COR | LI-COR: 925-32210; RRID:AB_2687825 | 1:10,000 |
| Antibody | Anti-rabbit IgG IRDye 680RD (donkey monoclonal) | LI-COR | LI-COR: 925-68073; RRID:AB_2716687 | 1:10,000 |
| Commercial assay or kit | Promega One-Glo EX Luciferase Assay System | Promega | Promega:E8110 | |
| Commercial assay or kit | Capsid ELISA | Progen | Progen:PRATV | |
| Software, algorithm | Python scripts for library design and data analysis | This paper | | https://github.com/churchlab/aav_rep_scan; *churchlab, 2023* |

## Library cloning

We generated *rep* variant libraries through pooled oligonucleotide (oligo) synthesis (Twist Biosciences) and subsequent Golden Gate Assembly using methods previously developed in our lab (*Ogden et al., 2019*). To begin, we generated two wild-type backbone plasmids, containing either the Rep78/68 open-reading frame or the *rep* and *cap* open-reading frames, and removed all BsaI, BbsI, EcoRV, SphI, and XbaI sites. Within the *rep* open-reading frame, we introduced a synonymous mutation at G339 (c.1017G>C). Within the VP open-reading frame, we introduced a synonymous mutation at V118 (c.354C>G) and a coding mutation F370Y (c.1109T>A). The *rep* gene was divided into 11 tiles and cloning for each tile was carried out separately. We designed 300-mer oligos to include 207 nucleotides of *rep*-coding sequence immediately followed by a BbsI site, an EcoRV site, another BbsI site, and a unique 20-nucleotide barcode sequence. All of these elements were flanked by BsaI and primer binding sites on either end of the synthesized sequence. Unique primer-binding sites were used for each tile. To enable cloning of both the Rep78/68 and WT AAV2 format libraries, oligos containing the Rep52/40 start codon (M225) were synthesized with and without the M225G mutation. All possible single-codon substitutions and deletions, including synonymous variants, were included in the library. All positions from the Rep78/68 start codon to the Rep78/52 stop codon (622 codons) were mutated. We did not include the eight codons at the end of the Rep68/40 open-reading frame in our mutagenesis as these positions overlap with the VP1 open-reading frame. Each codon variant was represented by at least two unique barcodes. For each tile, a minimum of 10 uniquely barcoded wild-type controls were included. The Rep78/68 and WT AAV2 libraries each had a total of 81,116 uniquely barcoded variants.

Following synthesis, we amplified the oligos for each tile (Q5 Hot Start High-Fidelity 2X Master Mix, NEB). In parallel, we amplified the backbone vector using primers that introduced BsaI sites. Vector PCR products were digested with BsaI-HF v2, DpnI, and recombinant shrimp alkaline phosphatase (rSAP, NEB) overnight and PCR purified (QIAGEN QIAQuick PCR Purification Kit) the following day. We then performed Golden Gate Assembly with the amplified oligos and vector PCR digest products (NEBridge Golden Gate Assembly Kit, BsaI-HF v2). Golden Gate Assembly reactions were cycled 100× (16°C for 5 minutes, 37°C for 5 minutes) and then heat inactivated. Golden Gate Assembly products were PCR-purified and eluted into 25 uL of Buffer EB (QIAGEN). Eluates were drop-dialyzed against 30 mL of water for 1 hr and transformed (Lucigen E. cloni 10G ELITE electrocompetent cells). Transformed cells were recovered in 1 mL Lucigen recovery media at 37°C for 1 hr and the entire volume of outgrowth was used to inoculate 50 mL LB + kanamycin cultures, which we grew at 30°C overnight. The following day, we midi-prepped these cultures via alkaline lysis (QIAGEN Plasmid Plus Midi-Prep Kit). This first cloning step enabled generation of intermediate products, which contained any wild-type *rep* sequence upstream of the mutated oligo followed by the mutant oligo. In this step, all wild-type sequences downstream of the mutant oligo were removed.

To reintroduce these missing wild-type sequences, we used a second round of Golden Gate Assembly and the internal BbsI sites present in each oligo. Before performing Golden Gate Assembly, we ran rolling circle amplification on our intermediate cloning products. Then, 10 ng of the intermediate plasmid products were incubated with 10 uM random hexamer primers and 1× phi29 DNA polymerase buffer (NEB) at 95°C for 3 min and then cooled to room temperature. We then mixed these samples with rolling circle amplification solution (10 uM random hexamers, 1× phi29 DNA polymerase buffer, 5 U phi29 DNA polymerase, 1 mM dNTPs, 2 mg/mL recombinant albumin, 0.02 U inorganic pyrophosphatase) at a 1:1 ratio and incubated them at 30°C overnight. The resulting rolling circle amplification products were directly digested with BbsI-HF, DpnI, and recombinant shrimp alkaline phosphatase (all NEB) at 37°C overnight. The following day, we ran the digest products on 1% Tris-acetate EDTA gels and extracted and purified the correctly sized products (QIAGEN QIAQuick Gel Extraction Kit). In parallel, we PCR-amplified the missing downstream wild-type sequences for each tile from the backbone vectors; the primers used here also added BbsI sites. We ran Golden Gate Assembly with the rolling circle amplification digest products and vector PCR products, BbsI-HF (NEB), and T4 DNA Ligase (NEB), cycling as described above. Golden Gate Assembly products were transformed and midi-prepped as above. This second cloning step resulted in plasmids containing a full-length *rep* gene with a single-codon mutation followed by a 20-nucleotide barcode. WT AAV2 library plasmids also contained a wild-type copy of the AAV2 *cap* gene between the variant *rep* genes and barcodes.

To enable packaging of variant *rep* sequences into AAV capsids, the step 2 cloning products were moved into inverted terminal repeat-containing vectors. Step 2 cloning products were subject to rolling circle amplification as above and digested with XbaI, SphI-HF, and EcoRV-HF. The *rep* and/or *cap* open-reading frames in the backbone vectors were flanked by XbaI and SphI sites. EcoRV sites were part of the synthesized oligos and should only be present in step 1 cloning products. Digest products were gel extracted and ligated into inverted terminal repeat-containing vectors. In the case of the WT AAV2 format library, the inverted terminal repeat vector contained the p5 promoter and endogenous AAV2 polyA sequence. In the case of the Rep78/68 and Rep52/40 libraries, the inverted terminal repeat vector contained a CMV promoter, WPRE, and bGH polyA sequence. The integrity of the inverted terminal repeat destination plasmids was confirmed by SmaI digest and complete plasmid sequencing (MGH DNA Core).

## Viral library production assay

HEK293T cells were cultured in DMEM supplemented with 10% FBS and seeded in five-layer cell culture multi-flasks (Corning 353144) at $8 \times 10^7$ cells/flask 2 d prior to transfection. We transfected HEK293T cells using polyethylenimine. For each library, replicate transfections were performed in separate cell stacks. For the pCMV-Rep78/68 library, transfections were performed with 1 ug of pCMV-Rep78/68 library plasmids, 50 ug of pHelper, 25 ug of pCMV-AAV2*cap*, and 1.5 ug of p19-Rep52/40 per cell stack. For the WT AAV2 library, transfections were performed with 1 ug of WT AAV2 library plasmids and 50 ug of pHelper. Additional plasmid ratios were tested for each library; conditions that resulted in the strongest correlation in production fitness values between replicate transfections were used; these are the conditions listed above. AAV5 and AAV9 capsid production assays were performed using the pCMV-Rep78/68 library as described above.

Three days post-transfection, NaCl was added to a final concentration of 0.5 M and samples were incubated at 37°C for 3 hr to detach and lyse cells. Samples were transferred to fresh 500 mL bottles and incubated at 4°C overnight. The following day, any precipitate was removed by pipetting and the remaining volume was passed through a sterile 0.22 um PES filter. PEG-8000 was added to a final concentration of 8% and samples were again incubated at 4°C overnight. The following day, we centrifuged samples at $3000 \times g$ for 20 min to pellet the PEG-precipitated virus. Supernatants were discarded and pellets resuspended in 8 mL of DPBS. We then digested samples with a 1:10,000 dilution benzonase (MilliporeSigma 1.101695.0001) at 37°C for 45 min and subjected them to iodixanol gradient ultracentrifugation as previously described (*Ogden et al., 2019*; *Zolotukhin et al., 1999*). Briefly, benzonase-treated samples were underlaid with an iodixanol gradient (MilliporeSigma D1556) in polypropylene tubes (Beckman Coulter, 362183) and centrifuged at $242,000 \times g$ for 1 hr at 16°C in a VTi50 rotor. Following ultracentrifugation, 500 uL fractions were collected from the 40% iodixanol layer and buffer-exchanged into DPBS using 100 kDa molecular weight cutoff centrifugal filter units (MilliporeSigma UFC910024).

*Rep* sequences in the purified pool represent variants capable of producing genome-containing viral particles. Barcodes from this viral pool, along with barcodes from the plasmid libraries, were amplified using flanking primer sequences. Illumina sequencing adapters and indices were added in a subsequent PCR. The resulting PCR amplicons were pooled and sequenced using the Illumina NextSeq platform (Biopolymers Facility, HMS). Barcode sequences were extracted from the resulting sequencing reads and barcode counts ($c_v$) from replicate transfections were summed. We calculated the frequency of each variant ($f_v$) in the viral library as $f_v = c/\Sigma c_v$ and then determined the production fitness for each variant as $s = f_{viral}/f_{plasmid}$. Finally, production fitness was normalized to that of the wild-type controls: $s = s/s_{WT}$.

## Determination of DNase-resistant particle titers by qPCR

To validate the results of our multiplexed library production assay, 14 single-codon *rep* variants were selected and individually transfected into HEK293T cells. Transfections were performed in triplicate. We seeded cells at $4 \times 10^5$ cells/well in six-well plates 24 hr prior to transfection. Small amounts of pCMV-Rep78/68 format variants, 50 pg per transfection, were used to recapitulate the low plasmid levels used in the library production assay. pCMV-Rep78/68 variants were transfected via PEI together with pHelper (2 ug), pCMV-AAV2*cap* (1 ug), and p19-Rep52/40 (2 ng) plasmids as in the library assay. Three days post-transfection, we lysed cells by 3× freeze–thaw. Samples were then clarified

by centrifugation at 15,000 × *g* for 5 min. Then, 5 uL of supernatant were subjected to DNase digest (Thermo Fisher, EN0521) at 37°C for 30 min followed by heat inactivation at 65°C for 10 min. We then incubated samples at 98°C for 10 min to denature the capsids and measured DNase-resistant particle titers with qPCR (IDT PrimeTime Gene Expression Master Mix). Primers and a probe binding to the 5′ region of the *rep* gene were used. The sequence of the forward primer was 5′-GAGCATCTGCCC GGCATTTC-3′, the sequence of the reverse primer was 5′-ATCTGGCGGCAACTCCCATT-3′, and the sequence of the probe was 5′-HEX-ACAGCTTTG-ZEN-TGAACTGGGTGGCCGA-3IABkFQ-3′ (IDT).

Transfections with *rep* variants cloned into pRepCap plasmids were performed as described above using the following amounts of plasmid for each transfection: 1 ug of pRepCap, 1 ug of inverted terminal repeat plasmid, and 2 ug of pHelper. The pEf1a-Luc-WPRE (ssAAV, Addgene plasmid #87951), pCAG-Luc (ssAAV, Addgene plasmid #83281), and pCAG-GFP (scAAV, Addgene plasmid #83279) inverted terminal repeat plasmids were a gift from Mark Kay (*Paulk et al., 2018*; *Pekrun et al., 2019*). We used qPCR to determine plasmid concentration for transfection. The *rep* primer and probe sequences mentioned above were used to quantify pRepCap plasmids. To determine the concentration of GFP-expressing plasmids, we used a forward primer with the sequence 5′-GAAC CGCATCGAGCTGAA-3′, a reverse primer with the sequence 5′-TGCTTGTCGGCCATGATATAG-3′, and a probe with the sequence 5′-FAM-ATCGACTTC-ZEN-AAGGAGGACGGCAAC-3IABKFQ-3′ (IDT). To determine the concentration of luciferase-expressing plasmids, we used a forward primer with the sequence 5′-CATGTACCGCTTCGAGGAG-3′, a reverse primer with the sequence 5′-GAAGCTAAATAG TGTGGGCACC-3′, and a probe with the sequence 5′-FAM-CTTGCGCAG-ZEN-CTTGCAAGACTA TAAGATTC-3IABKFQ-3′ (IDT). The same GFP- and luciferase-specific primers and probes were used to quantify the resulting rAAV vectors.

## Western blot analysis of VP and Rep protein levels

Following transfection, media were aspirated from cell culture plates and cells were washed with DPBS. We then lysed cells with 50 uL of lysis buffer per well (six-well plates). The lysis buffer contained 10 mM Tris–HCl (pH 8), 150 mM NaCl, 1% Triton X-100, and cOmplete mini protease inhibitor (one tablet/10 mL lysis buffer). Lysates were transferred to fresh 1.5 mL tubes and centrifuged at 15,000 × *g* for 5 min to clarify. Total protein concentrations were determined by Bradford assay (Thermo Scientific Pierce Coomassie Bradford Protein Assay Kit). We loaded 75 ug of total protein from each sample on 4–12% Bis–Tris gels (Thermo Fisher) and ran the gels for 45 min at 140 V. We then transferred the proteins to PVDF membranes (Thermo Fisher). Membranes were blocked with 5% milk in PBST for 1 hr at 4°C. Blots were then incubated with a 1:250 dilution of B1 anti-VP antibody or a 1:100 dilution of anti-Rep 303.9 antibody (both American Research Products) in blocking buffer overnight. Both primary antibody mixtures also included a 1:20,000 dilution of anti-β-actin antibody (Abcam). The following day, blots were washed 3× with PBST and incubated with secondary antibodies for 1 hr in blocking buffer with 0.01% SDS. Goat anti-mouse IRDye 800CW (LI-COR) and donkey anti-rabbit IRDye 680RD (LI-COR) secondary antibodies were used. Blots were washed 3× with PBST and 1× with PBS, and the near-infrared fluorescence of the secondary antibodies was visualized using the Sapphire imager.

## Analysis of stop codons in alternate reading frames

At each codon position, the average fitness value for all variants that introduced a premature stop codon into the +1 reading frame was determined. The average fitness value for all variants that did not introduce a stop codon into the +1 frame but were synonymous in the Rep frame to those that did was also calculated. Ten amino acid rolling averages for the +1 stops and +1 non-stops were calculated. These average fitness values were plotted and compared. The same procedure was used to determine the effect of premature stop codons in the +2 reading frame.

## Comparison of effects on synonymous and non-synonymous codons on production fitness

For this analysis, we used production fitness values (s′) calculated at the codon level. For each fitness value, we calculated two mean-centered fitness values. Positional mean-centered fitness values were calculated as $positional\ mean\ centered\ s' = s'_{codon}/average\ s'_{amino\ acid\ position}$. Synonymous codon mean-centered fitness values were calculated as $synonymous\ codon\ mean\ centered\ s' = s'_{codon}/average\ s'_{synonymous\ codon\ variants}$.

Methionine and tryptophan codons were excluded as these amino acids are coded for by only a single codon. The mean-centered fitness values for each codon variant were plotted versus amino acid position.

## Affinity purification of AAV2 capsids

We purified rAAV2 vectors produced with variant pRepCap plasmids with AVB Sepharose (Cytiva) using a previously reported protocol, which we modified for use with gravity columns (*Mietzsch et al., 2020*). A 4.4 kb inverted terminal repeat genome (pEf1a-Luc-WPRE, ssAAV, Addgene plasmid #87951) was used. pRepCap plasmids and the pEf1a-Luc-WPRE inverted terminal repeat plasmid were quantified by qPCR as discussed above. HEK293T cells were harvested 3 d post-transfection and lysed by 3× freeze–thaw. We then subject samples to benzonase digest as described above and centrifuged samples at $6000 \times g$ for 30 min to pellet cell debris. Samples were then passed through 0.2 um PES filters, diluted 1:1 with TD Buffer (DPBS, 1 mM $MgCl_2$, 2.5 mL KCl), and incubated with 4 mL of AVB Sepharose slurry at 4°C overnight with shaking. The following day, we loaded the samples onto gravity columns, washed with 20 mL of 1× TD Buffer, and eluted samples with 5 mL 0.1 M glycine–HCl (pH 2.6). Eluates were immediately neutralized with 500 uL of 1 M Tris–HCl (pH 10). All elution fractions were pooled and buffer-exchanged as above and samples volumes were normalized to 400 uL. rAAV viral genome titers were determined via qPCR using the same luciferase primer and probe sequences discussed above. Total particle titers were determined via capsid ELISA (Progen) according to the manufacturer's protocol. Transfections, affinity purifications, and titer determinations were performed in duplicate.

## Determination of relative transduction efficiency

Twenty-four hours prior to transduction, we seeded HEK293T cells at $2 \times 10^4$ cells/well in opaque white-bottom 96-well plates. Each rAAV prep was diluted 1:10,000 in serum-free media. Then, 100 uL of this dilution was added to each well of a 96-well plate. We performed eight transductions with each rAAV prep. Forty-eight hours after transduction, we lysed the cells and measured luciferase activity (Promega One-Glo EX Luciferase Assay System).

## Material, code, and data availability

The pCMV-Rep78/68 plasmid library created in this article is available through Addgene (RRID:Addgene_198050). The code and data for this article can be accessed at https://github.com/churchlab/aav_rep_scan.git (copy archived at *churchlab, 2023*). Illumina sequencing reads were uploaded to NCBI GEO (series accession number GSE226265).

# Acknowledgements

We thank Siddharth Iyer, Anna Maurer, Kaia Mattioli, Aleksandra Prochera, Takeyuki Miyawaki, and Erik Aznauryan for their feedback on this manuscript. We would also like to acknowledge members of the Biopolymers Facility at Harvard Medical School, including Taylor Fennelly, Ashley Ciulla Hurst, and Baldwin Dilone, for their assistance with library sequencing. This work was funded by the Synthetic Biology Platform at the Wyss Institute and the Wyss Institute faculty core allocation fund. Schematics were generated with BioRender.com and published using a CC-BY-NC-ND license with permission.

# Additional information

#### Competing interests

Nina K Jain, Pierce J Ogden: All authors are inventors on a patent application related to this work (provisional patent application 63/482,316). George M Church: A full list of GMC's tech transfer, advisory roles, and funding sources can be found on the lab's website: http://arep.med.harvard.edu/gmc/tech.html. All authors are inventors on a patent application related to this work (provisional patent application 63/482,316).

## Funding

| Funder | Grant reference number | Author |
|---|---|---|
| Wyss Institute Synthetic Biology Platform | | George M Church |

The funders had no role in study design, data collection and interpretation, or the decision to submit the work for publication.

## Author contributions

Nina K Jain, Conceptualization, Formal analysis, Investigation, Methodology, Writing – original draft, Writing – review and editing; Pierce J Ogden, Conceptualization, Software, Methodology, Writing – review and editing; George M Church, Conceptualization, Supervision, Funding acquisition, Writing – review and editing

## Author ORCIDs

Nina K Jain ⓘ https://orcid.org/0000-0003-3020-4162
Pierce J Ogden ⓘ https://orcid.org/0000-0003-3444-6214
George M Church ⓘ https://orcid.org/0000-0001-6232-9969

Reviewer #2 (Public Review): https://doi.org/10.7554/eLife.87730.3.sa1
Reviewer #3 (Public Review): https://doi.org/10.7554/eLife.87730.3.sa2
Author Response https://doi.org/10.7554/eLife.87730.3.sa3

# Additional files

## Supplementary files
• MDAR checklist

## Data availability

Illumina sequencing reads were uploaded to NCBI GEO (series accession number GSE226265). Code and analyzed data for this paper are available on GitHub (https://github.com/churchlab/aav_rep_scan. git copy archived at *churchlab, 2023*).

The following dataset was generated:

| Author(s) | Year | Dataset title | Dataset URL | Database and Identifier |
|---|---|---|---|---|
| Jain NK, Ogden PJ, Church GM | 2023 | Comprehensive mutagenesis of AAV2 rep | https://www.ncbi. nlm.nih.gov/geo/ query/acc.cgi?acc= GSE226265 | NCBI Gene Expression Omnibus, GSE226265 |

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
