## [Editor Report · eLife assessment]

This study presents a **valuable** and comprehensive mutagenesis map of the AAV2 *rep* gene, which will undoubtedly capture the interest of scientists working with adeno-associated viruses and those engaged in the field of gene therapy. The thorough characterization of massive *rep* variants across multiple AAV production systems bolsters the claims made in the study, highlighting its utility in enhancing our understanding of Rep protein function and advancing gene therapy applications. The evidence presented is **convincing** and establishes a strong foundation that will stimulate and inform future research in the field.

---

## [Referee Report · Reviewer #2 (Public Review)]

The authors use a high-throughput sequencing-based enrichment assay to measure how individual amino acids substitutions in the Rep proteins of AAV change the production of AAV. The key experiment involved the creation of all possible single codon mutations of the AAV2 rep gene in a barcoded format, transfection of the library into HEK293T cells for production of AAV, and sequencing to see which rep variants were enriched in the viral particles produced from the library. As the library rep variants were flanked by inverted terminal repeats for packaging into viral particles, the authors could use high-throughput sequencing of the barcodes to determine how much each rep variant supported the production of AAV. The rep gene libraries were cleverly made through a cloning process that ensured each mutant was attached to an exactly known 20nt barcode included in each mutagenic oligo (and subsequently moved to the end of the library genes by another cloning step). This allowed the authors to confidently observe nearly all rep variants in their experiments, resulting in a comprehensive map between Rep protein variants and AAV production. The overall map should act as a useful guide for AAV engineering. Not only did certain variants improve AAV production by ~2-fold and show generality across AAV capsid serotypes, the map might be used to predict greater effects through combinations of mutations, especially if augmented by natural evolutionary datasets and statistical learning.

In interpreting the results of this study, the reader should bear in mind that what has been measured and validated in high throughput is the production of intact genome-containing AAVs. The authors also successfully show transduction for selected high production variants. This is important as the efficiency by which an AAV preparations transduce cells is most relevant property for gene therapy.

Overall, this is a well-executed and well-analyzed study. The results support the conclusions and claims of the work. I see this work as a useful resource for engineering recombinant AAVs to increase their production, which should have broad impact as the use of AAVs in gene therapy grows.

---

## [Referee Report · Reviewer #3 (Public Review)]

The study by Jain et al. on recombinant adeno-associated viruses (rAAVs) represents a valuable contribution to the fields of virus genetics and gene therapy. As non-pathogenic vectors, rAAVs have become a popular choice for delivering gene therapies. The authors have previously investigated the effects of all possible single codon substitutions, deletions, and insertions in the AAV2 cap gene on AAV production. In this study, they extend their analysis to the AAV2 rep gene and rep genes in two additional capsid serotypes, establishing a genotype-phenotype landscape that enhances our understanding of Rep protein function and offers potential strategies for improving Rep function in gene therapy applications. The experimental design is rigorous, the analyses well-executed, and the interpretations of the data are convincing. While I have a few suggestions to further refine the study, I believe it is overall an excellent piece of research.

One aspect that may warrant further consideration is the assumption, as mentioned in Figure 2's legend, that synonymous mutations are neutral and can serve as controls for normalizing the production rate. However, Figures S5-6 and Figures S11-12 suggest that synonymous mutations are not necessarily neutral, as their distribution is similar to that of nonsynonymous mutations. Thus, it may be beneficial to more thoroughly examine the potential effects of synonymous mutations on the genotype-phenotype landscape.

Additionally, previous research by Jeff Collar and others has reported that synonymous mutations can affect mRNA levels through mRNA degradation rate. It would be interesting to determine if the 20-bp barcodes located at the 3' end are positioned within the untranslated regions and could thus be employed to quantify the mRNA levels of individual variants. This information could offer insight into another potential mechanism by which single codon mutations impact the production rate of rAAV.

The authors discovered several novel mutations that enhance AAV production yet are absent in natural occurrences. This intriguing finding could benefit from further elaboration, particularly with regard to the distribution of these mutations within the protein structure and the nature of the amino acid transitions involved. It would also be informative if the authors could provide a brief discussion as to why these mutations have not been observed in nature. For instance, could it be that optimal viral fitness necessitates an intermediate production rate rather than an excessively rapid one? Expanding on these points may further enrich the paper and offer valuable insights for readers.

The authors have taken commendable steps to address the concerns I raised in my previous evaluation. They have provided comprehensive clarifications, performed necessary revisions, and expanded upon certain key points in the manuscript.

---

## [Author Response]

The following is the authors’ response to the original reviews.

Thank you for overseeing the assessment of our manuscript, “Comprehensive mutagenesis maps the effect of all single codon mutations in the AAV2 rep gene on AAV production". We would also like to thank the reviewers for their feedback. We have carried out the suggested experiments that we feel are most central to our conclusions and summarized the revisions to the manuscript below.

We appreciate the reviewers’ suggestion with regards to testing different rAAV genomes. We have measured the effect of Rep variants on the production of rAAV containing three additional genomes: a 4.4 kb single-stranded genome, a 3.9 kb single-stranded genome, and a 2.1 kb self-complementary genome (Figures 5C and 5D). The DNase-resistant particles titers - reported as a percent of wild-type Rep titers - are relatively consistent across these three constructs as well as the 5.0 kb single-stranded genome previously tested.

We agree with the reviewers that measurement of the relative transduction efficiency of rAAV produced with different Rep variants is an important experiment to conduct. To address this, we transduced HEK293T cells with rAAVs, containing a luciferase genome, which were produced using two different Rep variants. When a constant volume of purified rAAV was used for transduction, we observed that the rAAV produced with the S110R Rep variant resulted in higher transduction than rAAV produced with wild-type Rep (as measured by luciferase signal). While we tested only a small number of variants, these results indicate that at least one of the Rep variants we identified can increase not only the viral genome titer but also the titer of transducing particles.

To generate this transduction data, we produced additional rAAV preps using S110R and Q439T Rep variants. In the previous version of this manuscript, we used the Q439T variant to produce rAAV and noted a 10% increase in the ratio of viral genomes: capsids as determined by comparison of qPCR and capsid ELISA titers. However, a similar increase was not observed in the more recent experiment discussed above. We attribute this discrepancy to changes in the plasmid quantification methods used for transfection. Previously, we quantified plasmids using a fluorometric assay (Qubit); in our more recent experiments, we used qPCR to quantify plasmids for transfection. qPCR provides a more accurate measurement of plasmid concentration due to the specific nature of the primers and probes used, which may account for the subtle shift in quantification. While outside the scope of the current work, it will also be interesting to further investigate the proportion of full capsids using additional Rep variants and more direct methods, such as cryoEM or analytical ultracentrifugation.

We agree with the reviewers’ observation that there are differences in the production fitness values for synonymous variants. However, the variation in production fitness values between synonymous variants is smaller than that between non-synonymous variants. We conducted the following analysis to clarify this point. We calculated two mean centered fitness values for each codon variant in the WT AAV2 library. The “positional mean centered fitness value” was determined using the production fitness values of all variants at a given amino acid position and describes how far a given fitness value diverges from the mean fitness value for that position. The “synonymous codon mean centered fitness value” was determined using the production fitness values of all synonymous variants at a given position and describes how far a given fitness value diverges from the mean fitness value for all its synonymous codon variants. We then plotted both mean centered fitness values versus amino acid position (Figure S8).

The distribution of mean centered selection values is narrower when calculated at the synonymous codon level as opposed to the position level. This indicates that, in general, synonymous variants have more tightly distributed production fitness values than non-synonymous variants. This observation precludes us from conducting a more thorough analysis of the effects of synonymous codons on AAV production. (Although, there is at least one instance where clear differences between synonymous codons can be observed (Figure S9C and Figure S9D).) We agree with the reviewers that synonymous variants almost certainly influence aspects of AAV production, such as genome replication, transcriptional regulation, mRNA stability, and protein expression. However, our assay measures the aggregate effect of rep variants on all steps in the AAV production process and is likely unable to detect the effects of synonymous variants on specific steps in this process if those steps are not rate-limiting. We have updated the discussion section to include an explanation of the above.

The X-axes in Figures 5B and 5D have been updated to plot s’ instead of percent WT titer. We have also added asterisks to indicate significance in Figures 5A and 5C. Thank you for these suggestions.

We agree with Reviewer 3 that it would be interesting to sequence barcodes from the mRNA pool. The 20 bp barcodes are located upstream of the polyA site and should be present in mRNA transcripts. Something to consider is that AAV2 transcripts expressed from all three promoters (p5, p19, and p40) are polyadenylated at the same site (Stutika et al., 2016). As such, in our WT AAV2 library, barcode representation in the mRNA pool would indicate the aggregate effect of a rep variant on the levels of all AAV2 transcripts. In the pCMV-Rep78/68 library, only two AAV2 transcripts are generated - a spliced and unspliced version of the p5 product. Sequencing of barcodes present in the mRNA pool could be informative regarding the effect of rep variants on combined Rep78/68 expression levels. However, we feel that this experiment is outside the scope of the current work.

We were also surprised at the number of novel functional Rep variants that were identified in our library. As the reviewer pointed out, optimal rAAV production likely does not equate to optimal fitness of naturally occurring AAV in the endogenous host. Naturally occurring AAV has both a latent and a lytic cycle and the Rep proteins play a role in both these processes (Pereira et al., 1997; Surosky et al., 1997). rAAV production, however, is primarily analogous to the lytic cycle of naturally occurring AAV. In their endogenous hosts, AAV must balance the effect of any mutations on fitness in both the lytic and latent contexts while we assay specifically for production fitness. We additionally attribute this finding to the relatively small number of AAV serotypes, for which rep sequences are available. We have added a discussion of the above to the manuscript.

Finally, in response to feedback from other researchers, we determined which amino acid substitutions resulted in production fitness values that were significantly different from that of wild-type (Figure S4). These results further emphasized the importance of the origin-binding domain; most statistically significant beneficial substitutions clustered here. Additionally, we noted that the majority of substitutions in the zinc-finger domain resulted in production fitness changes that were not significant. This lines up with previous work indicating that the zinc-finger domain is dispensable for rAAV production. We have added a discussion of these results to the main text.

We again thank the reviewers for their suggestions; we feel that incorporation of their suggestions has strengthened support for our conclusions and enhanced the utility of this work for others in the field.

ReferencesPereira, D. J., McCarty, D. M., & Muzyczka, N. (1997). The adeno-associated virus (AAV) Rep protein acts as both a repressor and an activator to regulate AAV transcription during a productive infection. Journal of Virology, 71(2), 1079–1088. https://doi.org/10.1128/jvi.71.2.1079-1088.1997

Stutika, C., Gogol-Döring, A., Botschen, L., Mietzsch, M., Weger, S., Feldkamp, M., Chen, W., & Heilbronn, R. (2016). A Comprehensive RNA Sequencing Analysis of the Adeno-Associated Virus (AAV) Type 2 Transcriptome Reveals Novel AAV Transcripts, Splice Variants, and Derived Proteins. Journal of Virology, 90(3), 1278–1289. https://doi.org/10.1128/JVI.02750-15

Surosky, R. T., Urabe, M., Godwin, S. G., McQuiston, S. A., Kurtzman, G. J., Ozawa, K., & Natsoulis, G. (1997). Adeno-associated virus Rep proteins target DNA sequences to a unique locus in the human genome. Journal of Virology, 71(10), 7951–7959. https://doi.org/10.1128/jvi.71.10.7951-7959.1997